# Interferon-Inducible Protein-10 as a Marker to Detect Latent Tuberculosis Infection in Patients with Inflammatory Rheumatic Diseases

**DOI:** 10.3390/jpm12071027

**Published:** 2022-06-23

**Authors:** Mediha Gonenc Ortakoylu, Ayse Bahadir, Sinem Iliaz, Derya Soy Bugdayci, Mehmet Atilla Uysal, Nurdan PAKER, Seda Tural Onur

**Affiliations:** 1Department of Pulmonary Disease, University of Health Sciences Turkey, Yedikule Chest Diseases and Thoracic Surgery Training and Research Hospital, Istanbul 34020, Turkey; aysebahadir@yahoo.com (A.B.); dratilla@yahoo.com (M.A.U.); sedatural@yahoo.com (S.T.O.); 2Department of Pulmonary Disease, Koc University Hospital, Istanbul 34010, Turkey; snmkaraosman@gmail.com; 3Department of Physical Medicine and Rehabilitation, Istanbul Physical Medicine and Rehabilitation Training Hospital, Istanbul 34186, Turkey; deryabugdayci@yahoo.com (D.S.B.); nurdanpaker@hotmail.com (N.P.)

**Keywords:** latent tuberculosis, inflammatory rheumatoid disease, interferon-inducible protein-10

## Abstract

It is important to identify cases of latent tuberculosis infection (LTBI) who are at risk for tuberculosis (TB) reactivation. We aimed to evaluate the performance of interferon (IFN)-gamma-inducible protein 10 (IP-10) as a marker to detect LTBI in patients with inflammatory rheumatic diseases (IRD). This study comprised 76 consecutive subjects with IRD. Patients with a history of TB or having active TB were excluded. In all patients, IP-10 level was measured and tuberculin skin test (TST) and QuantiFERON-TB Gold In-Tube test (QFT-GIT) were performed. Seventy patients with complete test results were analyzed. Twenty-one (30%) QFT-GIT-positive patients were defined as having LTBI. IP-10 yielded 2197 pg/mL cut-off point. At this cut-off point, IP-10 showed 89% specificity with a sensitivity of 91% (AUC: 0.950, 95% CI 0.906–0.994). TST, QFT-GIT, and IP-10 were positive in 77.1%, 30%, and 44.3% of the patients, respectively. Concordance among the results of TST, QFT-GIT, and IP-10 tests was evaluated. Agreement was poor between IP-10 and TST (58.6%, κ = 0.19), whereas it was good between QFT-GIT and IP-10 (84.3%, κ = 0.65). The results of the present study demonstrated that sensitivity and specificity of released IP-10 were as high as those of QFT-GIT in indicating LTBI in IRD patient group.

## 1. Introduction

Tuberculosis (TB) remains as an important problem all over the world. For a TB-free world, it is essential to reduce the prevalence of latent *Mycobacterium tuberculosis* infection and transition from latent infection to active disease [1]. Increase in immunosuppressive conditions highlight the need for additional strategies to maintain and improve TB control.

After the introduction of tumor necrosis factor (TNF)-α inhibitors (anti-TNFs) into inflammatory rheumatic diseases (IRD) treatment, we experienced more severe and more common TB infections in this group of patients [2,3]. Therefore, it has become obligatory to identify cases of latent tuberculosis infection (LTBI) prior to anti-TNFs [4].

Tuberculin skin test (TST), which is widely used in the diagnosis of LTBI, has some drawbacks, including variability in test application and low specificity due to purified protein derivative (PPD) presenting in non-tuberculous mycobacteria as well as in Bacille Calmette-Guérin (BCG) strains [5]. Moreover, TST use in IRD patients presents a major complicating factor: there is a decreased responsiveness of peripheral mononuclear cells, leading to a loss in delayed hypersensitivity, which is fundamental for the recognition of antigens, such as PPD [6].

Interferon (IFN)-gamma release assays (IGRAs) measure in-vitro T cell response to Mycobacterium TB-specific antigens [7]. Its results are more specific than those of TST, since mycobacterium TB-specific antigens [Early Secretory Antigenic Target-6 (ESAT-6) and Culture Filtrate Protein-10 (CFP-10)] that are not present in BCG and most of non-tuberculous mycobacteria are used during IGRA test on the contrary to PPD [8,9,10].

There are several unresolved issues on the potential clinical use of IGRAs. One area of controversy is whether they can be used in immunocompromised patients, in addition or as an alternative to the TST. The lack of a gold standard for LTBI diagnosis has complicated the assessment of diagnostic accuracy of IGRAs and the comparison of these tests with the TST [11]. Other Potential biomarkers to detect LTBI more accurately in IRD patients are needed. IFN-gamma-inducible protein-10 kDa (IP-10, CXCL10 [C-X-C motif chemokine 10]) is a pro-inflammatory chemokine involved in trafficking monocytes and T-cells to inflamed foci. In the studies, IP-10 level have been found to be much higher in the QuantiFERON-TB Gold in-Tube test (QFT-GIT) supernatants of TB patients as compared to the healthy individuals and it was thought that TB-specific antigen-stimulated IP-10 could be a potential biomarker for TB infection [12]. We aimed to evaluate the performance of IP-10 as a marker to detect LTBI and the agreement among TST or QFT-GIT in patients with IRD from moderate prevalence setting.

## 2. Materials and Methods

### 2.1. Study Population

This cross-sectional study comprised all consecutive subjects with IRD [rheumatoid arthritis (RA) and ankylosing spondylitis (AS)], who were being treated and followed in the Istanbul Physical Medicine and Rehabilitation Training and Research Hospital and referred to our pulmonology clinic to be evaluated in terms of pulmonary diseases during a six-month period of time. All patients’ BCG status and medications used for IRD were reviewed and their chest graphs were evaluated. Patients with a history of TB in the past and the patients considered to have active TB were excluded. All patients were Human immunodeficiency virus (HIV) (−). This study was approved by Yedikule Chest Diseases and Thoracic Surgery Training and Research Hospital Ethics Committee (ID:2017/ 73) and all patients read and signed the informed consent.

Of the patients, 28 were on TNF-α treatment, 12 were on steroid treatment, and 21 were on disease-modifying antirheumatic drugs (DMARD). We assessed BCG vaccination status based on interviewing past vaccinations and scar inspection. All subjects underwent a chest radiograph, they were questioned about history of previous TB, and results of previous TST.

### 2.2. Performing and Assessing TST

Immediately after blood was drawn, TST was performed by intradermal injection of 0.1 mL (5TU) of PPD (RT-23-tween80). The transverse diameter of induration was measured in millimeters 72 h later using the ballpoint pen method by only one examiner in all patients [13]. In IRD group, TST induration was interpreted according to published guideline as follows: 0 to 4 mm as negative and ≥5 mm as positive [14]. To maximize the detection rate for LTBI, two-step TST was performed, with a second TST administered 7–10 days after a negative initial test.

### 2.3. Whole Blood Stimulation

For the test, 1 mL of whole blood was drawn in each of three vacutainer tubes provided as part of the QFG-GIT system (Cellestis, Carnegie, Australia), a new generation of QFT test. These tubes are already pre-coated with saline (negative control), peptides of ESAT-6, CFP10, and antigen TB 7.7 (antigen stimulated), and PHA (positive mitogen control). The tubes were mixed and incubated for 20–24 h at 37 °C and they were frozen until further analysis. This plasma provided the unstimulated, antigen stimulated and mitogen stimulated supernatant samples.

### 2.4. IFN-Gamma Level Quantification

IFN-gamma measurement was performed by enzyme-linked immunosorbent assay (ELISA) using QFG-GIT test according to the manufacturer’s instructions [15]. A result of ≥0.35 IU/mL of IFN-gamma in the TB antigen tube minus the negative control (or nil) tube was considered a positive result. If the level was less than this and the mitogen control was positive (≥0.5 IU/mL), a negative result was recorded. If the level in both the TB antigen and mitogen tube was less than the threshold for positive, or the level in the nil tube was >8.0 IU/mL, then an indeterminate result was recorded. As there is no gold standard for the diagnosis of LTBI, and BCG being a routine in our vaccination programme, IRD patients with a positive QFT-GIT were defined as having LTBI.

### 2.5. IP-10 Level Quantification

For the quantification of IP-10 level, the RayBio Human IP-10 ELISA kit is used (RayBiotech, Inc., Norcross, GA, USA). The kit has a microplate with 96 wells pre-coated with a specific monoclonal antibody for human IP-10. IP-10 supernatant levels were measured using a sandwich ELISA according to the manufacturer’s instructions [16].

Samples were diluted 1:2. Fifty µL samples from each three QFG-GIT test tubes were taken and put in the wells. Wells were covered and incubated for 2.5 h at room temperature. After washing four times, 50 µL prepared biotin antibody was added to each well and incubated 1 h at room temperature. Washing procedure was repeated. A hundred µL streptavidin was added to the wells and incubated 45 min at room temperature. After the washing procedure, 50 µL of TMB One-Step Substrate Reagent was added to each well and incubate for 30 min at room temperature in the dark. Finally, 100 µL of Stop Solution was added to each well and read at 450 nm ELISA reader immediately.

### 2.6. Statistical Analysis

Qualitative measurements were defined in numbers and percentages. Descriptive analyses were expressed as means and standard deviation (SD) for normally distributed variables or median values and minimum-maximum for the non-normal distributed variables. Chi-square or Fisher’s exact test was used to compare frequencies or values of variables within the TST positive, QFT-GIT and IP-10 groups. We used receiver operating characteristic (ROC) analysis to examine discriminant validity. The concordance between TST, QFT-GIT and IP-10 was assessed by computing the Kappa statistics. Statistical analyses were performed using XLSTAT 2015.2.03 for Windows by Addinsoft. Statistical significance was at *p* < 0.05, two-tailed.

## 3. Results

A total of 70 subjects (51.4% males) with IRD were enrolled in the study. Out of 70 patients, 45 had AS and 25 had RA. The present population was characterized by a mean age of 47 ± 14 years; the prevalence of BCG was 77.1%. In IRD group, 54 (77.1%) patients had positive TST, 21 (30%) had positive QFT-GIT. Characteristics of the patients are demonstrated in Table 1.

### 3.1. Level of IP-10

TB antigen-stimulated plasma IP-10 level was found to be higher in QFT-GIT(+) patients (median: 26,060.3 ± 18,626.5 pg/mL, min–max: 165–60,000 pg/mL) as compared to QFT-GIT(−) patient group (median:1982.7 ± 3607.6 pg/mL, min–max: 0–20,982 pg/mL) (*p* < 0.0001). The released plasma IP-10 level was significantly higher in QFT-GIT(+) patients (median: 3680.5 ± 1954.0 pg/mL, min–max: 110–8899 pg/mL) as compared to QFT-GIT(−) patient group (median:789.5 ± 1543.7 pg/mL, min–max: 0–8115 pg/mL) (*p* < 0.0001).

### 3.2. Cut-Off Point Determination for IP-10

IP-10 value in nil tube was extracted from the plasma IP-10 value which was stimulated by TB antigen. Thus, we calculated the amount of IP-10 produced after stimulation with TB antigen. This value was called as released IP-10. To determine the diagnostic performance of TB antigen dependent IP-10, receiver-operator characteristic (ROC) curve analysis was performed in QFT-GIT(−) and QFT-GIT(+) IRD patients. It yielded 2197 pg/mL cut-off point. At this cut-off point, IP-10 showed 89% of specificity with a sensitivity of 91% (AUC: 0.950, 95% CI:0.906–0.994) (Figure 1).

Using the released IP-10, the results obtained with the cut-off value of ≥2197 pg/mL were considered positive. According to this cut-off value, IP-10 test was positive in 31 (44.3%) patients with IRD. Demographic and clinical features of the patients are demonstrated in Table 2.

Tuberculin skin test positivity was significantly higher in AS patients versus RA patients and IP-10 positivity was significantly higher in male versus female patients. No difference was determined between the patients with and without BCG vaccination, as well as the patients received and not received TNF-α inhibitor therapy, in terms of positivity of the tests (TST, QFT-GIT, IP-10). Likewise, it was observed that steroid use has no statistically significant effect on test results. TST positivity decreased with older age (p = 0.03). Contrary to TST, QFT-GIT and IP-10 positivity increased with age, but this increment was not statistically significant.

Concordance between the results of TST, QFT-GIT and IP-10 tests is demonstrated in Table 3. According to the TST threshold taken as ≥5 mm, agreement between QFT-GIT and TST was poor (55.7%, κ = 0.24), and it was also poor between IP-10 and TST (58.6%, κ = 0.19). The agreement was good between QFT-GIT and IP-10 (84.3%, κ = 0.65).

TST induration was observed in IRD patients with TST(+)/QFT-GIT(+) results (16.09 ± 4.35 mm); and with TST(+)/QFT-GIT(−) results (14.09 ± 4.20 mm) (*p* = 0.123). Likewise, the diameter of TST induration was found to be 16.11 ± 4.37 mm in TST(+)/IP-10(+) IRD patients and 13.18 ± 4.53 mm in TST(+)/IP-10(−) patients (*p* = 0.06).

## 4. Discussion

We aimed to evaluate the performance of IP-10 as a marker to detect LTBI and the concordance among TST or QFT-GIT in patients with IRD. The results revealed good agreement between QFT-GIT and IP-10 but poor agreement between TST and either QFT-GIT or IP-10.

It has been demonstrated that TST had poor performance and gave false negative results in the diagnosis of LTBI in immunocompromised patient group including those with IRD [6]. In a study that evaluated TST response between RA patients and healthy controls, TST was found to be negative independent from duration and activity of disease in 76.6% of the patients with RA, whereas it was negative in 26% of the control group [17]. In the present study, TST positivity was found to be 77% in IRD patient group (86.1% in males, 67.6% in females). BCG scar was present in 82.4% of the patients. In a large-scale study performed in normal population living in a tuberculosis endemic country, TST positivity (TST ≥ 10 mm) was found to be 69.3%. At least one BCG scar was detected in 91.2% of the cases and it was demonstrated that TST positivity is higher in male gender (70% vs. 55%) [18]. Similarity of these results with the results of present study suggests that TST responsiveness is not poor at all in our study group. Since there is no gold standard in the diagnosis of LTBI, all studies on IRD patient group in the literature are based on evaluating and comparing TST and new-generation IGRA tests and investigating agreement between these tests. In the IRD patient group, IGRA positivity was reported to be 13–44%, whereas TST positivity was reported to be 1–61%. Different inflammatory diseases of the patients in the study populations and different immunosuppressive medications that they have been receiving and also different BCG vaccination status in their own populations make the interpretation of outcomes difficult [19]. In a large meta-analysis, the sensitivity and specificity of IGRA tests were found to be 76% and 98%, respectively. It has been stated that IGRA tests have excellent specificity that is not influenced by BCG vaccination but that data from pediatric and immunocompromised patients are limited [8]. In the studies conducted in IRD patient group, the agreement between TST and IGRA was found to be good in the countries where the rate of BCG vaccination is low, whereas it was found to be poor in the countries where the rate of vaccination is high [10,20]. In the present study, the agreement between TST and QFT-GIT was poor (kappa = 0.24).

An important IGRA-related disadvantage is high indeterminate test results in immunocompromised patients. It was reported as 13–38.2% depending on the patient group and it was emphasized that this was accompanied by immunosuppressive therapy and particularly peripheral lymphocytopenia [21,22]. This is different in the IRD patient group; it was reported that indeterminate IGRA results were not prevalent in IRD patients and were seen by less than 5% [19]. The prevalence of indeterminate test results was reported to be 1.2% in an IRD patient group consisted of 398 patients [11]. Among 70 patients in our study, there was no patient with indeterminate IGRA result.

In 2006, QFT-GIT supernatants were screened for cytokines and chemokines that could be the markers of *M. tuberculosis* antigen-specific cell-mediated immune response. It was demonstrated that IP-10 was overexpressed in the patients with active tuberculosis but there was no expression in unexposed controls. This suggested that IP-10 might be an alternative marker to IFN-γ and may lead to development of IGRA tests [23]. Studies were performed to investigate diagnostic performance of IP-10 in tuberculosis and they have been frequently compared with IGRA tests. Studies that compared the results of IP-10 and QFT-GIT one-to-one in indicating active TB reported comparable rates of positivity (74–91% vs. 79–100%) [12,24,25]. Consistency between the two tests was investigated in only two of these studies and was found good. Specificity was found to be 98% for IP-10 and 100% for QFT-GIT in unexposed healthy controls in Italy and Denmark and no cross-reaction was observed with BCG [26]. IP-10 was found to be 55% positive and QFT-GIT was found to be 48% positive in an Indian population without symptom or contact with tuberculosis and it was interpreted in the way that the test could reflect the prevalence of LTBI [27]. In a multicenter study conducted in Europe in the patients with non-tuberculosis disease, IP-10 was 35% positive and QFT-GIT test was 27% positive and increase in the positivity of IP-10 was interpreted in the way that IP-10 might have superior sensitivity or impaired specificity as compared to QFT-GIT [25]. Obtaining similar results in the present study (IP-10 31% positive, QFT-GIT 21% positive) suggested that sensitivity of IP-10 might be considered to be superior. Also, a few recent studies showed that IP-10 might help to differentiate healthy controls, LTBI, and active TB from each other, but it did not reach sufficient sensitivity and specificity [28,29].

Making accurate diagnosis of LTBI is important for the treatment and follow-up in immunocompromised patients, who have high risk of progression to active TB. In a novel randomized controlled study, the occurrence of tuberculosis was evaluated in patients receiving TNF-α inhibitors and exposure to TNF-α inhibitors was associated with a statistically significant threefold increase in the risk of TB. These findings confirm that appropriate screening with TST/IGRA test should be performed before starting treatment with TNF-α inhibitors [30]. Murdaca et al. also pointed to the negative TST should be interpreted with caution in any patient who is under treatment with an immunosuppressive agent as they ae more likely to have false-negative TST results, IGRAs for the LTBI have recently been proven to be more spesifiic for LTBI than the TST in immunocomponent subjects as an expert opinion [31].

Unfortunately, both TST and IGRA tests may give false negative outcomes in such patients. In a study conducted with IP-10 in HIV(+) patient group, it was reported that IP-10 is slightly better than QFT-GIT in demonstrating active TB and is influenced less by CD4 cell count [25]. In another study by Villar-Hernandez et al., using IP-10 and IFN-γ together led to higher detection rate of LTBI in patients with inflammatory rheumatic diseases without being affected by immunosuppressive treatment [32].

A study, which evaluated IP-10 in the diagnosis of LTBI in RA patients receiving immunosuppressive therapy, reported significantly higher IP-10 level in TST(+) patients; similar results were obtained in the present study. In the same study, it was demonstrated that baseline unstimulated IP-10 level was high but decreased after treatment in the patient that developed active TB [33]. Serum IP-10 measurement appears to be one of the most promising tests that might contribute to the diagnosis of LTBI.

The most striking problem in the comparison of studies that are conducted with IP-10 as a new diagnostic marker is associated with technical aspects of the measures used to detect IP-10. The use of different brand kits in the studies, measuring the samples at different dilutions, wide range of measures, and taking different cut-off values in the studies have been specified as the main problems [34]. Detecting cut-off value using released IP-10 narrowed the range of measure in our study. Limitations of the present study, as in the similar studies, include the facts that study population is small, since it is a specific patient group, and patients have been receiving different therapies. However, the limited number of studies investigating IP-10 in a specific patient group makes the present study critical.

In conclusion, the results of the present study demonstrated that IP-10 is as comparable as QFT-GIT in the diagnosis of LTBI in IRD patient group. It suggests that IP-10 assay could be an alternative biomarker for diagnosis of LTBI in countries where the BCG vaccine is routinely administered.

## Figures and Tables

**Figure 1 jpm-12-01027-f001:**
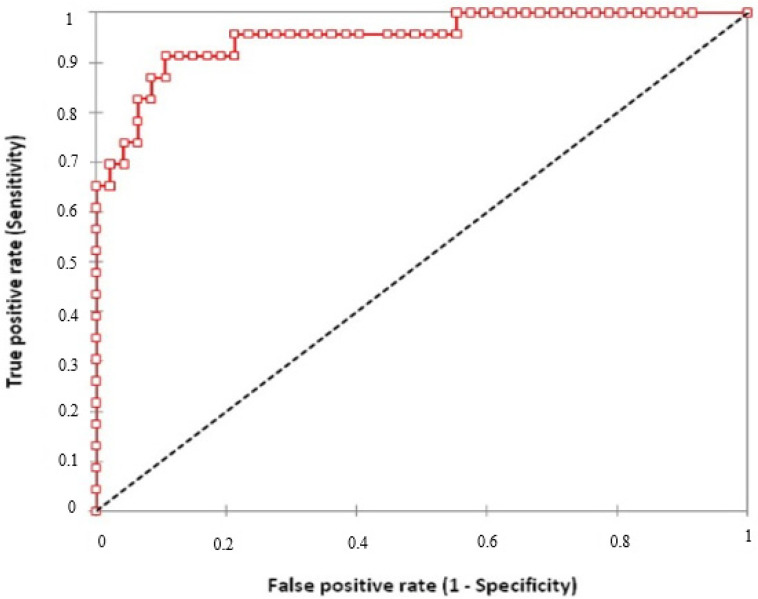
ROC curve analysis for the released plasma inducible protein-10. The cut-off values were determined using QFT-GIT(+) and QFT-GIT(−) patients with inflammatory rheumatic diseases. The area under curve was 0.950 (95% CI 0.906–0.994). (Red line: stimulated IP-10 levels; Black line: reference).

**Table 1 jpm-12-01027-t001:** Characteristics of the patients (*n* = 70).

Type of IRD ^1^	(*n*/%)
Ankylosing spondylitis, *n*/%	45/64.3%
Rheumotoid arthritis, *n*/%	25/35.7%
Mean age, years ± SD	47 ± 14
Male, *n*/%	36/51.4%
BCG positivity, *n*/%	54/77.1%
TST positivity, *n*/%	54/77.1%
QFT-GIT positivity, *n*/%	21/30%

^1^ IRD, inflammatory rheumatic diseases; BCG, Bacillus Calmette–Guérin, TST, tuberculin skin test; QFT-GIT, QuantiFERON-TB Gold in-Tube test.

**Table 2 jpm-12-01027-t002:** Associations of demographic, epidemiological and clinical characteristics and therapy with QuantiFERON-TB Gold intube (QFT-GIT) and tuberculin skin test (TST) and Interferon-Inducible Protein-10 (IP-10) positive results in subjects with IRD patients.

	*n*	TST (+) ^1^ (*n* = 54)	*p*	QFT-GIT (+) (*n* = 21)	*p*	IP-10 (+) (*n* = 31)	*p*
Gender, *n* (%)			0.660				
Female	34	23 (67.6)		8 (23.5)	>0.05	9 (26.5)	0.004
Male	36	31 (86.1)		13 (36.1)	data	22 (61.1)	
Age, years, *n* (%)			0.03 -		>0.05		0.66
<29	8	7 (87.5)		1 (12.5)		2 (25.0)	-
30–49	35	31 (88.6)		10 (28.6)		16 (45.7)	
50–69	22	15 (68.2)		10 (45.5)	data	11 (50.0)	
>70	5	1 (20.0)		0 (0.0)	data	2 (40.0)	
BCG vaccinated, *n* (%)							
Yes	54	44 (82.4)	0.230	15 (42.9)	0.190	22 (43.1)	0.160
No	16	10 (64.3)		6 (29.4)	data	9 (64.3)	
Diagnosis, *n* (%)							
RA	25	14 (56.0)	0.020	6 (24.0)	0.410	9 (12.9)	0.290
AS	45	40 (88.8)		15 (33.3)		22 (31.4)	
Treatment TNF-α inhibitors, *n* (%)							
Yes	28	20 (71.4)	0.350	5 (25.0)	0.700	9 (12.9)	0.090
No	42	34 (80.9)		16 (38.0)		22 (52.3)	
Steroid, *n* (%)							
Yes	12	8 (75)	0.45	5 (62.5)	0.25	6 (50)	0.52
No	58	46 (85.2)		16 (27.6)		25(43.1)	

^1^ TST assessment limit according to ≥5 mm. TST, tuberculin skin test; QFT-GIT, QuantiFERON-TB Gold In-Tube test; IP-10, interferon-gamma-inducible protein 10; BCG, Bacillus Calmette–Guérin; RA, rheumatoid arthritis; AS, ankylosing spondylitis; TNF, tumor necrosis factor.

**Table 3 jpm-12-01027-t003:** Concordance among latent tuberculosis infection tests.

Tests		Test Results			Kappa Value
	−/−	+/−	−/+	+/+	
TST/QFT-GIT	16	33	0	21	0.24
TST/IP-10	12	27	4	27	0.19
QFT-GIT/IP-10	38	1	11	20	0.65

TST: Tuberculin skin test, QFT-GIT: QuantiFERON-TB Gold In-Tube test, IP-10: Interferon-gamma-inducible protein 10. TST assessment limit according to ≥5 mm.

## Data Availability

The data presented in this study are available on request from the corresponding author. The data are not publicly available due to privacy reasons.

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
