# Peer review of "Interferon-Inducible Protein-10 as a Marker to Detect Latent Tuberculosis Infection in Patients with Inflammatory Rheumatic Diseases"

_jpm, 2022, doi:10.3390/jpm12071027_

Round 1

Reviewer 1 Report

The authors presented the results of testing IP-10 as a potential LTBI marker.

The methodology raises the main reservation, because the results do not refer to the PCR test with Mycobacterium tuberculosis, but other LTBI serological markers, the limitations of which are given in the text by the authors themselves.

According to the exclusion criteria (lines 71-72) patients with IRD were excluded. Is. it true? Please clarify.

The importance of IP-10 in LTBI diagnostics has already been studied in many studies - citations 25-32. What does this work add to knowledge and / or practice? What is the novelty of this work?

Author Response

Dear Sir /Madam,

First of all I would like to thank you for your comments, opinions and your suggestions. Please find my responses below.

Regarding your opinions about methodology;

There is no gold standard test for the diagnosis of LTBI. It is not possible to detect Mycobacterium Tuberculosis by PCR or other bacteriological methods in LTBI. Tuberculin skin test (TST) and/or Interferon-gamma release assays (IGRAs) are used for the diagnosis of LTBI. Our study was designed to compare the tests used in the diagnosis of LTBI with the IP-10 test and to investigate the compatibility.

In our study, it was investigated whether the IP-10 test could be used in the diagnosis of LTBI in patients with inflammatory rheumatoid diseases (IRD) and compared with TST and IGRAs, and as a result it was shown that it gave similar results with IGRAs. Since it is not a gold standard test used in the diagnosis of LTBI, it is not possible to say that this test is better, but it can be said that IP-10 test is as sensitive as IGRAs in the IRD patient group in countries where the BCG vaccine is routinely administered. The result of the study shows that the IP-10 test should be considered in the diagnosis of LTBI.

The incomprehensibility in the sentence “According to the exclusion criteria (lines 71-72) patients with IRD were excluded.” has been corrected. Thank you very much for your contribution.

Regarding your opinions about novelty;

In the IRD group, where anti-TNF alpha therapy is widely used, there is a worldwide consensus currently recommended in guidelines on investigating LTBI before starting anti-TNF therapy.

TST used in the diagnosis of LTBI may result as false negative or false positive with the effect of BCG vaccine. Although it has been reported that IGRAs are more sensitive in this patient group, false negative results can be obtained.

It has been shown that the IP-10 test may be an alternative biomarker in the diagnosis of LTBI for the IRD patient group in countries with a BCG vaccination policy, and we think that this will contribute to the literature and practice. This output has been added to the conclusion following your opinion.

Reviewer 2 Report

The paper is interesting and well written. I suggest to discuss the impact of the risk of latent TBC reactivation in patients with rheumatic diseases receiving TNF alpha inhibitors (see and add as references papers by Murdaca et al concerning thsi topic)

Author Response

Dear Sir /Madam,

First of all I would like to thank you for your comments, opinions and your suggestions. Please find my responses below.

Regarding your suggestion on adding the concerning article of Murdaca et al. as references;

I have read the regarding articles of Murdaca et al. and added them as references into the manuscript. Thank you very much again for your valuable suggestion.

Regarding your comment about “moderate English changes required”;

In order to give you a quick response and also because it is not a extensive English revisions comment, we have checked the manuscript again but due to time constraints it is not revised professionally or by a native English-speaking colleague.